# Beyond Unidirectional Flow: LLM Reasoning with Bidirectional Cycle-Consistent CoT

## Abstract

Small-large model collaboration is a promising approach for efficient reasoning, where lightweight assistant models generate intermediate representations to guide larger, more capable models. However, this paradigm encounters two key challenges: **representation heterogeneity** between different model architectures and **unidirectional information flow** that prevents mutual learning. Small assistant models and large base models develop distinct geometric structures for encoding similar concepts, making direct alignment difficult and leading to information degradation. Additionally, unidirectional flow creates asymmetric dynamics where assistant models cannot benefit from large models' superior representational capacity. We introduce **CycleCoT**, a bidirectional framework that addresses these bottlenecks through cycle-consistent soft thought alignment. Our approach uses dual residual transformation networks to establish invertible mappings between heterogeneous model spaces through three mechanisms: (1) expressive mappings between different model representations, (2) bidirectional alignment objectives enforcing semantic consistency in both directions, and (3) cycle consistency constraints preserving information during round-trip transformations. This enables large models' knowledge to enhance assistant models' soft thought generation, creating symbiotic collaboration. Evaluation on LLaMA-3.1-8B-Instruct and Qwen2.5-7B-Instruct across mathematical, commonsense, and symbolic reasoning benchmarks demonstrates consistent improvements over unidirectional baselines, with gains up to $5.5\%$ on mathematical reasoning tasks. Our analysis reveals that alignment quality surpasses quantity: fewer, well-aligned soft thoughts outperform longer sequences. Code is available at https://anonymous.4open.science/r/CycleCoT-0B7D/.

## 1 Introduction

Large Language Models (LLMs) excel at complex reasoning tasks but require prohibitive computational resources for practical deployment (Brown et al., 2020; OpenAI, 2023; Hoffmann et al., 2022; Narayanan et al., 2021). This cost makes it difficult to serve full-scale models at interactive latency. Small-large model collaboration offers a practical approach, where a lightweight assistant runs ahead of the main model to draft continuations, filter search branches, or supply domain-specific priors, so the large model can be more focused on high-difficult reasoning tasks (Kim et al., 2023; Leviathan et al., 2023; Chen et al., 2023). Since the assistant is inexpensive to adapt, the paradigm also enables rapid task personalization while reusing a frozen, general-purpose large base model (Xu et al., 2025).

Chain-of-Thought (CoT) reasoning exemplifies this collaborative paradigm, where small assistant models generate intermediate reasoning steps to guide large models toward improved solutions (Wei et al., 2022; Kojima et al., 2022). While traditional CoT operates through discrete token generation, recent work has explored continuous representations to overcome tokenization bottlenecks (Zhang et al., 2025; Hao et al., 2024). These continuous approaches enable small assistant models to generate latent "soft thoughts" that steer large models through learned projection mechanisms (Xu et al., 2025; Wang et al., 2025), offering computational advantages over sequential token-based reasoning. However, current frameworks face fundamental architectural limitations that constrain their collaborative effectiveness.

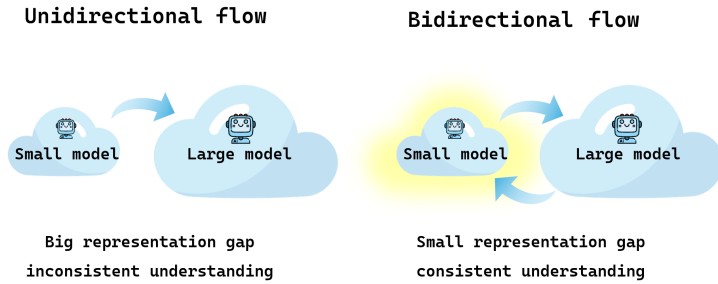

Figure 1: **Unidirectional vs. Bidirectional Collaboration. Left:** Conventional unidirectional flow from a small to a large model results in a large representation gap and inconsistent understanding. **Right:** Our proposed bidirectional flow enables mutual learning, reducing the representation gap and aligning the models for consistent, and synergistic reasoning.

**Challenges.** The core challenge lies in the following aspects. **(1) Representation heterogeneity** emerges as small assistant models and large capable models develop distinct geometric structures for encoding similar concepts, making direct alignment fundamentally difficult (Jiao et al., 2020; Sun et al., 2019; Zhang et al., 2018). Existing projection mechanisms struggle with this mismatch, leading to information degradation and suboptimal reasoning guidance. Additionally, **(2) single directional information flow** creates an asymmetric learning dynamic where the assistant model cannot benefit from the large model superior representational capacity, fundamentally limiting the collaborative potential of the framework.

**Proposed Method**. We introduce CycleCoT, a bidirectional framework that addresses these limitations. Our approach replaces unidirectional projectors with dual residual transformation networks that establish cycle-consistent mappings between small assistant and large model representation spaces. This bidirectional architecture enables the large model's contextual knowledge to flow back and enhance the assistant model's soft thought generation, creating a symbiotic learning dynamic. A comprison between unidirectional and bidirectional flow are shown in the Figure 1. Technically, CycleCoT operates through three mechanisms: (i) dual residual transformation networks that provide expressive mappings between heterogeneous model spaces, (ii) bidirectional alignment objectives that enforce semantic consistency across representations, and (iii) cycle consistency constraints that preserve information during round-trip transformations. Importantly, during the inference time, CycleCoT requires only the forward projector, maintaining computational efficiency while delivering improved soft thought quality (He et al., 2016; Zhu et al., 2017). To summarize, our contributions are threefold:

- *Fundamental Bottleneck Identification.* This study identifies and formalizes representation heterogeneity and single directional information flow as fundamental bottlenecks that limit the effectiveness of current learning paradigm in small and large model collaboration.
- *Bidirectional Alignment Framework.* The proposed CycleCoT introduces dual residual transformation networks with bidirectional alignment and cycle consistency objectives to achieve semantic alignment, revealing that heterogeneous representation spaces require learnable mappings and that small models can transcend capacity limitations through feedback from larger models.
- *Superior Empirical Performance.* Extensive experiments demonstrate consistent and significant improvements across mathematical, commonsense, and symbolic reasoning benchmarks, showing that semantically aligned soft thoughts substantially outperform both discrete CoT chains and existing continuous CoT baselines.

## 2 RELATED WORK

Our work is related to the following key research areas, including continuous reasoning, small-large model collaboration and bidirectional and cycle-consistent alignment.

**Reasoning with Continuous Thoughts**. Chain-of-Thought (CoT) prompting has significantly improved the reasoning capabilities of LLMs by eliciting explicit, intermediate steps (Wei et al., 2022; Kojima et al., 2022). The performance of CoT often scales with the length of the reasoning chain (Fu et al., 2022), but this incurs substantial computational costs from generating and processing long token sequences. To address this efficiency bottleneck, research has diverged into two main directions. The first direction aims to improve the quality and efficiency of the discrete reasoning process itself (Mohtashami et al., 2023). This includes methods that employ tree-based search to explore and verify multiple reasoning paths, such as Tree of Thoughts (Yao et al., 2023) and CoCoNUT (Hao et al., 2024). The second, more radical direction is to bypass discrete tokens entirely, proposing a paradigm shift towards continuous, latent-space reasoning (Wu et al., 2025). This approach materializes the reasoning process directly within the model's latent space, using "soft thought" vectors as implicit reasoning steps (Xu et al., 2025; Cheng & Van Durme, 2024).Our work, CycleCoT, builds upon this paradigm by generating latent "soft thoughts" to steer large models efficiently.

**Small-large Model Collaboration**. Within the continuous reasoning paradigm, a promising strategy to balance reasoning performance with computational efficiency is multi-model collaboration, where a smaller, adaptable "assistant" model assists a larger, powerful "base" model (Chen et al., 2024). Current approaches can be broadly categorized. One line of work focuses on inference acceleration, using the small model to generate draft outputs that the large model can efficiently verify, as seen in speculative decoding (Kim et al., 2023). Another direction involves task decomposition, where the small model acts as a planner that breaks down complex tasks for the large model to execute (Song et al., 2023). A third paradigm, which we focus on, is latent guidance, where the assistant generates continuous "soft thoughts" to steer the base model's reasoning process (Xu et al., 2025). However, this latent guidance approach is fundamentally constrained by the unidirectional information flow and representation heterogeneity between the two models. Our work aims to resolve these core bottlenecks.

**Bidirectional and Cycle-Consistent Alignment**. Our solution is also inspired by aligning different representational spaces. For instance, dual learning has shown that enforcing two-way consistency can benefit related tasks (Wang et al., 2024), and cycle consistency has proven to be a powerful regularizer for learning robust, information-preserving mappings between unpaired domains, famously used in image translation (Zhu et al., 2017). While these techniques are powerful, they have not yet been adapted to resolve the aforementioned challenges of representation heterogeneity and unidirectional flow in latent-space LLM collaboration. Our work is the first to bridge this gap, establishing a symbiotic, co-evolutionary learning dynamic between the assistant and base models.

## 3 METHODOLOGY

Our proposed method has three core components: (1) **Soft Thought Generation**: We leverage a frozen, small assistant language model to generate a compact sequence of instance-specific latent prompts, termed *soft token*, tailored to the input question. (2) **Bidirectional Representation Alignment**: We introduce a novel *Residual Projector* to bridge the representation and dimensionality gap between the assistant and the base LLM. This projector establishes a robust and approximately invertible alignment between their respective latent spaces. (3) **Latent-Conditioned Reasoning and Training**: The projected soft thoughts are integrated into the input of the main base LLM, which then autoregressively generates the reasoning chain and final answer. The system is trained end-to-end with a composite objective combining a standard language modeling loss with bidirectional alignment and cycle consistency regularizers to enforce representational fidelity.

### 3.1 SOFT THOUGHT GENERATION

**CoT Reasoning.** Given a question sequence $Q = [q_1, \ldots, q_{|Q|}]$, CoT reasoning decomposes the prediction task into two sequential steps: (i) generating a rationale sequence $R = [r_1, \ldots, r_{|R|}]$ and (ii) producing the final answer $A = [a_1, \ldots, a_{|A|}]$. The autoregressive generation process follows:

$$r_{i+1} = \text{LLM}(Q; R_{\leq i}), \tag{1}$$

$$a_{j+1} = \text{LLM}(Q; R; A_{\leq j}), \tag{2}$$

where $R_{\leq i} = [r_1, \ldots, r_i]$ and $A_{\leq j} = [a_1, \ldots, a_j]$ denote the partial sequences up to positions $i$ and $j$, respectively. While classical CoT operates in the discrete token space, recent approaches propose

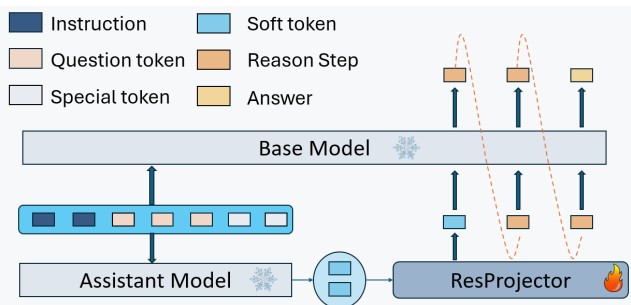

Figure 2: **Training and Inference Pipeline.** The frozen assistant model generates $N$ soft-thought slots. These are projected into the base LLM's hidden space and injected at designated thought positions to guide reasoning. Training utilizes bidirectional alignment and cycle consistency losses, while inference only requires the efficient forward projector, preserving low latency.

reasoning in the continuous (latent) space to circumvent tokenization bottlenecks (Chen et al., 2025; Hao et al., 2024).

In this work , we introduce an auxiliary small assistant LLM (frozen) to produce instance-specific *soft token* representations for $N$ placeholder slots inspired by (Xu et al., 2025). The small assistant model input is constructed as:

$$Query = \text{concat}\big[I_{\text{assist}}, Q, [\text{UNK}]_{1:N}\big], \tag{3}$$

where $I_{\text{assist}}$ is a task-specific instruction template, $Q$ is the input question sequence, and $[\text{UNK}]_{1:N}$ represents $N$ special placeholder tokens as shown in Figure 2. Feeding $Query$ into the small assistant model $LLM_A$ yields the final-layer hidden states: $H^{(\text{assist})} = LLM_A(Query)$, where $T_a$ is the sequence length and $d_a$ is the hidden dimension of the assistant model. From these hidden states, we extract the $N$ soft-token vectors corresponding to the placeholder positions: $T_A = H^{(\text{assist})}_{s_a:e_a}$, where indices $(s_a, e_a)$ specify the start and end positions of the $N$ placeholder tokens in the assistant model's sequence. This *assistant-produced* latent sequence serves as a compact, instance-specific soft prompt for downstream reasoning.

### 3.2 BIDIRECTIONAL REPRESENTATION ALIGNMENT

To bridge the representation and dimensionality gap between the assistant model ($d_a$) and the base LLM ($d_b$), prior work typically projects $T_A$ unidirectionally into the large base model space (assistant→base). However, such unidirectional projection can accumulate representation mismatch and drift over long reasoning chains (Li et al., 2023). To address this limitation, we propose a **Residual Projector** to map representations between the two spaces:

$$\Phi_{A \to B} : \mathbb{R}^{d_a} \to \mathbb{R}^{d_b}, \quad \Phi_{B \to A} : \mathbb{R}^{d_b} \to \mathbb{R}^{d_a}. \tag{4}$$

Each projector is implemented as a two-layer MLP with a residual connection and Layer Normalization. When $d_a \neq d_b$, a linear projection is applied to the skip connection to align dimensions. Forward mapping injects soft thoughts into the base LLM space: $T_{A'} = \Phi_{A \to B}(T_A)$, while reverse mapping is employed during training for reconstruction constraints and alignment regularization, establishing an approximately invertible alignment between the two representation spaces as shown in Figure 3.

### 3.3 TRAINING OBJECTIVES

Our training objective combines the primary language modeling loss with two auxiliary alignment regularizers to ensure consistent representation alignment.

**Language Modeling Loss.** We apply standard autoregressive negative log-likelihood on the ground-truth CoT trajectories and answers:

$$\mathcal{L}_{\text{LM}} = -\sum_{t \in \mathcal{I}_R \cup \mathcal{I}_A} \log p(y_t | y_{<t}, x_{\text{LLM}}), \tag{5}$$

Figure 3: **Framework of proposed method.** The assistant model's ($LLM_A$) soft thoughts ($T_A$) are mapped to the base model's space via the forward projector $\Phi_{A\to B}$. The reverse projector $\Phi_{B\to A}$ maps representations back to the assistant's space. This enables bidirectional alignment and a cycle consistency loss on the round-trip transformation ($T_{A''}$), which reduces the representation gap.

where $\mathcal{I}_R$ and $\mathcal{I}_A$ index the rationale and answer tokens respectively, $y_t$ represents the target token at position $t$, and $y_{<t}$ denotes all preceding tokens (Brown et al., 2020). The tokens preceding the thought region are masked during loss computation to prevent label leakage.

**Bidirectional Alignment Loss.** On the thought segment, we enforce bidirectional MSE alignment between the assistant and base model representations. The bidirectional alignment loss is defined as:

$$\mathcal{L}_{\text{align}} = \frac{1}{2} \left\| T_{A'} - T_B \right\|_2^2 + \frac{1}{2} \left\| T_{B'} - T_A \right\|_2^2. \tag{6}$$

This loss encourages the projected representations from both directions to align with their respective target hidden states (Zhang et al., 2018).

**Cycle Consistency Loss.** To prevent long-horizon representation drift and encourage approximate invertibility of the projectors, we impose cycle consistency over the thought segment. This loss combines MSE reconstruction error with cosine similarity preservation:

$$T_{A''} = \Phi_{B\to A} \left( \Phi_{A\to B}(T_A) \right), \tag{7}$$

$$\mathcal{L}_{\text{cycle}} = \left\| T_A - T_{A''} \right\|_2^2 + \left( 1 - \cos(T_A, T_{A''}) \right), \tag{}$$

where $\cos(\cdot, \cdot)$ computes the cosine similarity between vectors (Zhu et al., 2017). In practice, we slice the thought spans for each sample within the batch, accumulate losses across both directions, and compute the average.

**Overall Training Objective.** The complete training objective combines all loss components:

$$\mathcal{L} = \mathcal{L}_{\text{LM}} + \lambda_{\text{align}} \cdot \mathcal{L}_{\text{align}} + \lambda_{\text{cycle}} \cdot \mathcal{L}_{\text{cycle}}, \tag{8}$$

where $\lambda_{\text{align}}$ and $\lambda_{\text{cycle}}$ are hyperparameters controlling the relative importance of the alignment and cycle consistency losses, respectively.

### 3.4 Latent-Conditioned Reasoning and Training

The base LLM input is constructed by concatenating the task instruction, question, and projected soft thoughts: $x_{\text{LLM}} = \text{concat}\left[ I_{\text{LLM}}, Q, T_{A\to B} \right]$, where $I_{\text{LLM}}$ is a task-specific instruction template for the base model and $T_B$ are the $N$ soft-thought vectors projected by $\Phi_{A\to B}$ (Li & Liang, 2021). For compatibility with standard autoregressive decoding, we directly replace the input embeddings at the designated thought slots with $T_B$, allowing the base LLM ($LLM_B$) to generate intermediate reasoning and the final answer:

$$\tilde{R} = LLM_B(x_{\text{LLM}}), \tag{9}$$

$$\tilde{A} = LLM_B(x_{\text{LLM}}, \tilde{R}). \tag{10}$$

Table 1: Dataset characteristics for CycleCoT evaluation

| Dataset | Domain | Training | Test | Format |
|---------|--------|----------|------|--------|
| GSM8K | | 7,473 | 1,319 | Generation |
| ASDiv-Aug | Mathematical | 4,183 | 1,038 | Generation |
| AQuA | | 97,467 | 254 | Multiple Choice |
| StrategyQA | Commonsense | 1,832 | 458 | Binary |
| CommonsenseQA | | 9,741 | 1,221 | Multiple Choice |
| DU | Symbolic | – | 369 | Multiple Choice |

During training, we supervise $\tilde{R}$ and $\tilde{A}$ using standard language modeling objectives. At inference, only the forward projector $\Phi_{A \rightarrow B}$ is utilized, maintaining computational efficiency comparable to SoftCoT.

**Computational Complexity**   Let $N$ denote the number of thought slots and $d = \max(d_a, d_b)$. During training, the bidirectional projector introduces $O(Nd^2)$ additional computation for the two MLPs across the thought span. However, the language modeling cost typically dominates for longer output sequences. At inference, only $\Phi_{A \rightarrow B}$ is used once to produce $T_B$, yielding identical asymptotic complexity to unidirectional SoftCoT approaches.

## 4 EXPERIMENTAL EVALUATION

In this section, we aim to address these research questions:

- **RQ1:** How does CycleCoT's bidirectional alignment framework perform compared to existing approaches across diverse reasoning domains?

- **RQ2:** How do the individual components of CycleCoT contribute to overall performance improvements?

- **RQ3:** How effectively does bidirectional alignment bridge the representation gap between different LLMs?

- **RQ4:** How does CycleCoT balance computational efficiency with reasoning performance across different token configurations?

### 4.1 EXPERIMENTAL SETUP

**Model Configuration.** We evaluate CycleCoT using two representative model pairs from 2 LLM families. The base models are LLaMA-3.1-8B-Instruct (Dubey et al., 2024) and Qwen2.5-7B-Instruct (Yang et al., 2024), serving as primary reasoning engines. The assistant models are LLaMA-3.2-1B-Instruct and Qwen2.5-1B-Instruct, respectively, responsible for generating contextual soft representations. This configuration enables assessment of cross-architectural generalizability.

**Benchmark Datasets.** Our evaluation encompasses six reasoning benchmarks across three cognitive domains: Mathematical Reasoning: GSM8K (Cobbe et al., 2021), ASDiv-Aug (Xu et al., 2025), and AQuA (Ling et al., 2017). Commonsense Reasoning: StrategyQA (Geva et al., 2021) and CommonsenseQA (Talmor et al., 2018). Symbolic Reasoning: Date Understanding (DU) (Srivastava et al., 2023) for temporal reasoning evaluation. Table 1 summarizes the dataset characteristics and evaluation protocols.

**Baseline Methods.** We compare CycleCoT against four representative baseline approaches. LoRA Fine-Tuning (Hu et al., 2022) results are reported from (Xu et al., 2025) for direct comparison. Coconut (Hao et al., 2024) and SoftCoT (Xu et al., 2025) are implemented using their official source code to ensure fair evaluation. Zero-Shot CoT is evaluated by directly prompting the respective base models (LLaMA-3.1-8B-Instruct and Qwen2.5-7B-Instruct) with instructions without additional training. To ensure statistical reliability, we conduct five independent runs with different random seeds in our main experiments (Coconut, SoftCoT, Zero-Shot CoT, and CycleCoT), and report mean performance with standard deviations across all runs.

Table 2: Performance comparison on LLaMA-3.1-8B-Instruct across reasoning domains

| Method | GSM8K | ASDiv-Aug | AQuA | StrategyQA | CommonsenseQA | DU | Avg. |
|---|---|---|---|---|---|---|---|
| *LLaMA-3.1-8B-Instruct* | | *Mathematical* | | | *Commonsense* | *Symbolic* | |
| LoRA Fine-Tuning | $75.66 \pm 0.00$ | $86.67 \pm 0.00$ | $52.36 \pm 0.00$ | – | – | – | – |
| Coconut | $76.88 \pm 0.36$ | $86.24 \pm 0.72$ | $52.98 \pm 0.82$ | – | $73.42 \pm 0.94$ | – | – |
| Zero-Shot CoT | $79.88 \pm 0.43$ | $86.84 \pm 0.96$ | $54.85 \pm 1.18$ | $65.49 \pm 2.08$ | $73.42 \pm 0.94$ | $54.98 \pm 2.29$ | 69.06 |
| SoftCoT | $79.27 \pm 0.31$ | $86.79 \pm 0.38$ | $54.59 \pm 2.07$ | $65.11 \pm 1.28$ | $74.23 \pm 1.06$ | $58.18 \pm 2.67$ | 69.70 |
| **CycleCoT** | $\mathbf{85.37} \pm 0.65$ | $\mathbf{87.67} \pm 0.26$ | $\mathbf{56.29} \pm 0.85$ | $\mathbf{67.78} \pm 2.02$ | $\mathbf{74.34} \pm 0.16$ | $\mathbf{60.56} \pm 2.10$ | **72.00** |

Table 3: Performance comparison on Qwen2.5-7B-Instruct across reasoning domains

| Method | GSM8K | ASDiv-Aug | AQuA | StrategyQA | CommonsenseQA | DU | Avg. |
|---|---|---|---|---|---|---|---|
| *Qwen2.5-7B-Instruct* | | *Mathematical* | | | *Commonsense* | *Symbolic* | |
| LoRA Fine-Tuning | $81.80 \pm 0.00$ | $86.80 \pm 0.00$ | $62.60 \pm 0.00$ | – | – | – | – |
| Coconut | $82.45 \pm 0.77$ | $87.22 \pm 0.46$ | $63.61 \pm 0.59$ | – | $73.74 \pm 0.25$ | – | – |
| Zero-Shot CoT | $82.46 \pm 0.81$ | $88.09 \pm 0.34$ | $65.09 \pm 2.29$ | $50.39 \pm 2.21$ | $72.96 \pm 0.44$ | $65.41 \pm 1.78$ | 70.73 |
| SoftCoT | $84.47 \pm 1.26$ | $87.79 \pm 0.85$ | $70.34 \pm 1.22$ | $60.16 \pm 2.47$ | $\mathbf{74.85} \pm 0.22$ | $\mathbf{66.17} \pm 2.47$ | 73.96 |
| **CycleCoT** | $\mathbf{86.44} \pm 0.92$ | $\mathbf{89.85} \pm 0.19$ | $\mathbf{73.49} \pm 1.29$ | $\mathbf{64.59} \pm 3.95$ | $73.95 \pm 0.36$ | $65.59 \pm 3.01$ | **75.65** |

## 4.2 MAIN RESULTS

**How CycleCoT Performs Across Diverse Reasoning Domains. (RQ1)** Our method CycleCoT achieves superior performance across both different base models and tasks, demonstrating consistent improvements over existing approaches. The method reaches 72.00% average accuracy on LLaMA-3.1-8B-Instruct and 75.65% on Qwen2.5-7B-Instruct, substantially outperforming various baseline methods. In the following, we present the detailed findings.

For mathematical reasoning, the results (Tables 2 and 3) reveal a compelling pattern where Cycle-CoT's bidirectional alignment particularly excels at complex mathematical reasoning tasks. The dramatic improvement on GSM8K (+6.1% for LLaMA) suggests that the cycle consistency mechanism helps preserve crucial numerical relationships and multi-step logical dependencies that are often lost in unidirectional projection. Interestingly, Qwen models show their strongest gains on AQuA (+3.15%), a dataset requiring abstract algebraic reasoning, indicating that the bidirectional alignment better captures the semantic relationships between different mathematical concepts across model representations.

For commonsense and symbolic reasoning, while mathematical tasks benefit from preserving precise numerical relationships, commonsense reasoning reveals a different advantage of our approach. As shown in Tables 2 and 3, the substantial improvements on StrategyQA (+2.67% for LLaMA, +4.43% for Qwen) demonstrate that bidirectional alignment helps transfer the nuanced contextual understanding that smaller assistant models possess. For the DU dataset, which lacks a training set, we directly evaluate using models trained on GSM8K, demonstrating effective cross-domain transfer from mathematical to symbolic reasoning tasks. This suggests that the cycle consistency constraint prevents the degradation of implicit world knowledge during cross-model projection, enabling more effective reasoning about everyday scenarios and temporal relationships.

For cross-dataset knowledge transfer, to assess whether CycleCoT learns generalizable reasoning patterns rather than dataset-specific artifacts, we evaluate transfer performance within the mathematical reasoning domain by training models on one dataset and testing on another. As shown in Table 4, cross-dataset training often matches or even exceeds in-dataset performance, with cross-trained models achieving within 0.5% of their in-domain counterparts. This suggests that the cycle consistency constraint forces the model to learn more abstract mathematical reasoning representations that transfer across different datasets within the same domain. Similar transfer patterns are observed for Qwen models (Table 6). For cross-dataset evaluation, we further report results averaged over three random seeds to ensure robustness.

Table 4: Cross-dataset generalization on LLaMA-3.1-8B-Instruct

| Training | Testing | Accuracy | vs Zero-shot | vs In-domain |
|----------|---------|----------|--------------|--------------|
| – | | $54.85 \pm 1.18$ | – | $-1.44$ |
| AQuA | AQuA | $56.29 \pm 0.85$ | $+1.44$ | – |
| ASDiv-Aug | | $56.07 \pm 2.23$ | $+1.22$ | $-0.22$ |
| – | | $86.84 \pm 0.96$ | – | $-0.83$ |
| ASDiv-Aug | ASDiv-Aug | $87.67 \pm 0.26$ | $+0.83$ | – |
| AQuA | | $87.19 \pm 1.11$ | $+0.35$ | $-0.48$ |

Table 5: Ablation studies on Qwen2.5-7B-Instruct

| Configuration | GSM8K | ASDiv-Aug | AQuA | StrategyQA |
|---------------|-------|-----------|------|------------|
| *Qwen2.5-7B-Instruct* | | *Mathematical* | | *Commonsense* |
| **CycleCoT (Full)** | $\mathbf{86.44} \pm 0.92$ | $\mathbf{89.85} \pm 0.19$ | $\mathbf{73.49} \pm 1.29$ | $\mathbf{64.59} \pm 3.95$ |
| w/o Residual Projector | $83.16 \pm 1.58$ | $86.39 \pm 0.73$ | $70.58 \pm 1.24$ | $54.49 \pm 1.28$ |
| w/o Bidirectional Losses | $80.58 \pm 1.12$ | $86.24 \pm 2.13$ | $69.78 \pm 0.67$ | $59.21 \pm 1.87$ |
| w/o Cycle Loss | $83.20 \pm 0.85$ | $88.15 \pm 0.52$ | $71.80 \pm 1.15$ | $62.30 \pm 1.85$ |
| w/o Align Loss | $84.15 \pm 0.75$ | $87.40 \pm 0.65$ | $72.10 \pm 1.05$ | $61.85 \pm 2.25$ |

## 4.3 ABLATION STUDIES

**How CycleCoT Components Contribute to Performance Gains. (RQ2)** The ablation study (Table 5) reveals several key insights about CycleCoT's design. Residual projectors provide substantial improvements, particularly on StrategyQA ($+10.10$ points), while removing bidirectional losses causes significant performance drops across all tasks. However, a counter-intuitive phenomenon emerges: when residual projectors are used without bidirectional losses, performance often falls below simpler configurations, suggesting that sophisticated projectors without proper constraints can introduce harmful representational distortions. The individual loss function analysis demonstrates that both cycle consistency and alignment losses contribute meaningfully to performance—removing cycle consistency drops GSM8K to $83.20\%$ while removing alignment loss yields $84.15\%$, both representing clear degradation compared to the full model's $86.44\%$. Similar ablation patterns are observed on LLaMA models, with detailed results provided in Appendix A.4. All ablation experiments are further averaged over three random seeds to ensure robustness.

## 4.4 ANALYSIS

**How Bidirectional Alignment Bridges Representation Gaps (RQ3)** The most compelling evidence for CycleCoT's effectiveness lies in how it fundamentally transforms cross-model representation alignment. Figure 4 demonstrates that while traditional unidirectional approaches struggle with representation drift and semantic inconsistency, our bidirectional framework achieves remarkable alignment improvements across multiple geometric measures. The $69.3\%$ reduction in Euclidean distance reveals that projected representations become substantially closer to their target spaces, while the $19.1\%$ improvement in cosine similarity demonstrates better preservation of semantic directionality. Most striking is the $90.1\%$ reduction in mean squared error, indicating that the cycle consistency constraint virtually eliminates point-wise projection errors that plague linear approaches.

This dramatic improvement suggests that bidirectional alignment doesn't merely reduce noise—it fundamentally restructures the representation space to maintain semantic coherence across model boundaries. The geometric analysis reveals several key insights: First, the substantial Euclidean distance reduction indicates that our dual projectors learn more accurate mappings between heterogeneous spaces compared to simple linear transformations. This suggests that the representation gap between different model architectures is not merely a scaling or rotation issue, but requires sophisticated non-linear transformations to bridge effectively. Second, the preserved cosine similarity demonstrates that bidirectional alignment maintains the angular relationships between concept vectors, which is crucial for preserving semantic structure during cross-model transfer. Third, the near-elimination of mean squared error through cycle consistency reveals that information loss during

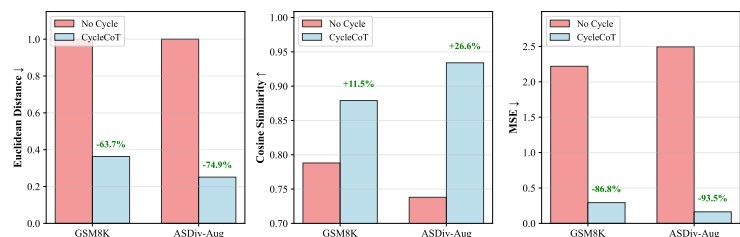

Figure 4: Representation alignment comparison: CycleCoT vs. No Cycle

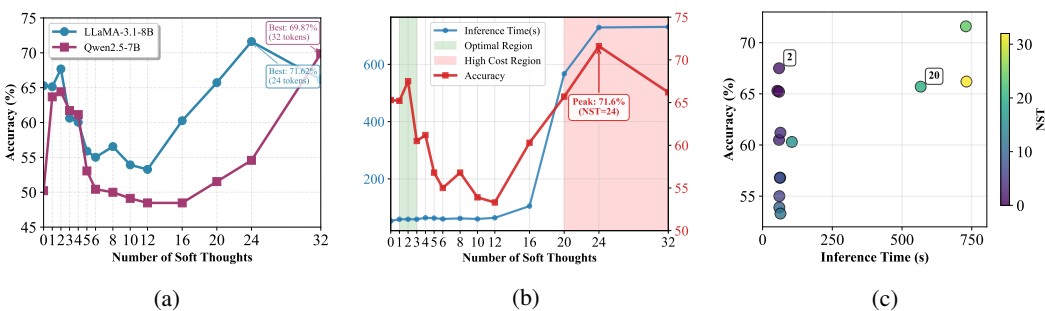

(a)                                    (b)                                    (c)

Figure 5: CycleCoT computational efficiency analysis (a) Performance vs. soft thoughts count (b) Cost vs. performance trade-off (c) Performance vs. cost scatter plot

round-trip transformations—a critical failure mode in unidirectional approaches—can be effectively mitigated through explicit invertibility constraints. These findings collectively indicate that successful cross-model alignment requires both forward accuracy and backward consistency, validating our core hypothesis that bidirectional constraints are essential for robust representation transfer.

**How Efficiency-Performance Trade-offs Are Analyzed. (RQ4)** The relationship between soft token quantity and performance reveals a counterintuitive principle that challenges conventional wisdom about representation capacity (Figures 5a, 5b, and 5c). Rather than following a monotonic "more is better" pattern, both model families exhibit a distinctive U-shaped performance curve that exposes the critical importance of alignment quality over raw quantity. Peak performance occurs with minimal tokens $(1 - 4)$, demonstrating that well-aligned representations can capture complex reasoning patterns efficiently. The computational analysis reveals three distinct operational regimes: the efficient regime (NST $\leq 5$) maintains stable inference times while achieving strong performance, making it ideal for real-time applications; the transition regime (NST $6 - 16$) shows exponential cost growth with minimal accuracy gains, representing a computational "dead zone" that should be avoided; and the high-performance regime (NST $> 16$) achieves peak accuracy but requires dramatically increased resources. The emergence of NST = 2 as the optimal configuration—achieving $67.5\%$ accuracy in just $59$ seconds—validates CycleCoT's core design philosophy: sophisticated alignment mechanisms enable strong performance with minimal computational overhead.

## 5 CONCLUSION

We presented CycleCoT, a bidirectionally aligned, consistency-regularized framework for continuous-space reasoning that addresses the core bottlenecks of representation gap and limited assistant capacity in latent CoT. Our approach leverages a frozen small assistant model to generate soft thoughts, which are then mapped between latent spaces using a novel Residual Projector. This projector is trained with a composite objective, combining a language modeling loss with a bidirectional alignment loss to enforce representational similarity and a cycle consistency loss to ensure the mapping is approximately invertible. Together, these mechanisms transform the base model from a passive consumer of thoughts into an active source of reverse guidance, creating a symbiotic partnership that demonstrably improves reasoning performance.

## ETHICS STATEMENT

This work adheres to ethical principles in AI research. We exclusively utilize publicly available and widely accepted benchmark datasets that do not contain personally identifiable information or sensitive content. We acknowledge that the underlying large language models used in this research may inherit biases from their training data, and our methodology does not specifically address or mitigate these inherent biases. We advocate for responsible deployment practices, including human oversight and careful validation of AI-generated content, when applying our framework or similar technologies in real-world scenarios. Furthermore, we are committed to transparency and reproducibility by detailing our methods and intend to make our code publicly available to facilitate further research and scrutiny. While we acknowledge the environmental impact of training large AI models, we prioritize computational efficiency where feasible and encourage continued efforts to reduce the carbon footprint of AI research.

## REPRODUCIBILITY STATEMENT

To ensure full reproducibility of our findings, all source code for the CycleCoT framework, trained model checkpoints for our projectors, and the configuration files used for all experiments will be made publicly available upon acceptance. An anonymized repository is provided for review at: `https://anonymous.4open.science/r/CycleCoT-0B7D/`. Our experiments are conducted on publicly accessible benchmarks, including GSM8K, ASDiv-Aug, and StrategyQA, and build upon standard open-source models such as LLaMA-3.1-8B-Instruct and Qwen2.5-7B-Instruct, all available via the Hugging Face Hub. A comprehensive list of hyperparameters, compute requirements, and specific dataset pre-processing steps are detailed in Appendix. All released artifacts will comply with their original open-source licenses, and data sources will be clearly acknowledged.

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

---

**Algorithm 1:** Training with Bidirectional Alignment and Cycle Consistency

---

**Input:** Assistant model $\mathcal{A}$, Base LLM $\mathcal{B}$, projector $\Phi_{A \to B}, \Phi_{B \to A}$, weights $(\lambda_{\text{align}}, \lambda_{\text{cycle}})$,
      indices $(s_a, e_a), (s_b, e_b)$.

**for** *minibatch* $\{(x_{\text{assist}}, x_{\text{LLM}}, y)\}_{b=1}^{B}$ **do**

    $H^{(\text{assist})} \leftarrow \mathcal{A}(x_{\text{assist}})$ ;                              `// final hidden states`

    $T_A \leftarrow H^{(\text{assist})}_{s_a:e_a}$ ;                                 `// N soft thoughts`

    $T_B \leftarrow \Phi_{A \to B}(T_A)$

    Replace base input embeddings on $(s_b : e_b)$ with $T_B$;

    $\text{out} \leftarrow \mathcal{B}(x_{\text{LLM}})$;

    $L_{\text{LM}} \leftarrow \text{NLL}(\text{out.logits}, y)$;

    $H^{(\text{base})} \leftarrow \text{out.hidden\_states}[-1]$;

    $H^{(\text{base})}_N \leftarrow H^{(\text{base})}_{s_b:e_b}$;

    $L_{\text{a2b}} \leftarrow \frac{1}{N} \| \Phi_{A \to B}(T_A) - H^{(\text{base})}_N \|^2_F$;

    $L_{\text{b2a}} \leftarrow \frac{1}{N} \| \Phi_{B \to A}(H^{(\text{base})}_N) - T_A \|^2_F$;

    $L_{\text{align}} \leftarrow \frac{1}{2}(L_{\text{a2b}} + L_{\text{b2a}})$;

    $T''_A \leftarrow \Phi_{B \to A}(\Phi_{A \to B}(T_A))$;

    $L_{\text{cycle}} \leftarrow \frac{1}{N} \| T''_A - T_A \|^2_F + (1 - \cos(T''_A, T_A))$;

    $L \leftarrow L_{\text{LM}} + \lambda_{\text{align}} L_{\text{align}} + \lambda_{\text{cycle}} L_{\text{cycle}}$;

    Update $\Phi_{A \to B}, \Phi_{B \to A}$ (and optional adapters) by backprop on $L$;

**end**

---

# A   Appendix

## A.1   Algorithm

## A.2   Implementation and Training Details

We implement CycleCoT with 4 soft thoughts as the standard configuration across most datasets and models. The training hyperparameters are optimized for different model architectures: for LLaMA models, we set alignment loss weight $\lambda_{\text{align}} = 0.15$ and cycle consistency weight $\lambda_{\text{cycle}} = 0.08$; for Qwen models, we use $\lambda_{\text{align}} = 0.10$ and $\lambda_{\text{cycle}} = 0.05$ to account for architectural differences.

The BiDirectional Projector employs residual MLP blocks with bottleneck size 1024, ReLU activation, and LayerNorm for all model configurations. Training is conducted for up to 10 epochs using AdamW optimizer with learning rate $2 \times 10^{-5}$ and batch size 8(4 for AQuA dataset). We implement early stopping mechanisms to conserve computational resources and improve training efficiency. All training experiments are performed on an NVIDIA H800 GPU, while inference is conducted on an NVIDIA RTX 5090 GPU with mixed precision (bfloat16) for optimal efficiency.

During inference, the small model generates 4 soft thoughts which are then projected to the large model's representation space through our bidirectional alignment framework. This configuration provides an optimal balance between reasoning capability and computational cost across diverse reasoning tasks. More details can be found in our anonymous GitHub repository.

## A.3   Additional Cross-Dataset Results

Table 6 presents cross-dataset generalization results for Qwen2.5-7B-Instruct, complementing the LLaMA results shown in the main text.

## A.4   Additional Ablation Results

Table 7 presents ablation study results for LLaMA-3.1-8B-Instruct, complementing the Qwen results shown in the main text.

Table 6: Cross-dataset generalization on Qwen2.5-7B-Instruct

| Training | Testing | Accuracy | vs Zero-shot | vs In-domain |
|----------|---------|----------|--------------|--------------|
| – | | $65.09 \pm 2.29$ | – | $-8.40$ |
| AQuA | AQuA | $73.49 \pm 1.29$ | $+8.40$ | – |
| ASDiv-Aug | | $73.62 \pm 1.80$ | $+8.53$ | $+0.13$ |
| – | | $88.09 \pm 0.34$ | – | $-1.76$ |
| ASDiv-Aug | ASDiv-Aug | $89.85 \pm 0.19$ | $+1.76$ | – |
| AQuA | | $87.96 \pm 0.91$ | $-0.13$ | $-1.89$ |

Table 7: Ablation study on LLaMA-3.1-8B-Instruct

| Configuration | GSM8K | ASDiv-Aug | AQuA | StrategyQA |
|---------------|-------|-----------|------|------------|
| *LLaMA-3.1-8B-Instruct* | | *Mathematical* | | *Commonsense* |
| **CycleCoT (Full)** | $\mathbf{85.37} \pm 0.65$ | $\mathbf{87.67} \pm 0.26$ | $\mathbf{56.29} \pm 0.85$ | $\mathbf{67.78} \pm 2.02$ |
| w/o Residual Projector | $81.96 \pm 1.46$ | $86.08 \pm 3.09$ | $54.63 \pm 1.87$ | $67.93 \pm 3.09$ |
| w/o Bidirectional Losses | $80.20 \pm 0.33$ | $86.27 \pm 3.47$ | $52.21 \pm 1.26$ | $55.10 \pm 1.28$ |

## A.5 LLM USAGE STATEMENT

In accordance with ICLR policy, we report on the use of a large language model (Google's Gemini) as an assistive tool in the preparation of this manuscript. The LLM was utilized for drafting, iteratively refining, and stylistically polishing several sections, including the Abstract, Related Work, and Conclusion. It also assisted in the literature review process and the formatting of academic references. The core scientific contributions—encompassing the initial research ideation for the Cycle-CoT framework, experimental design, and the execution and analysis of all results—were conceived and conducted exclusively by the human authors. The authors directed all stages of research, validated all LLM-generated content, and assume full responsibility for the final claims and conclusions of this work.

