# OpenReview forum: "Beyond Unidirectional Flow: LLM Reasoning with Bidirectional Cycle-Consistent CoT"
_ICLR.cc/2026/Conference — ICLR 2026 Conference Withdrawn Submission_

### Official Review · Reviewer_sZkQ · 2025-10-16

**Soundness:** 2
**Presentation:** 2
**Contribution:** 2
**Rating:** 2
**Confidence:** 4

**Summary:**

This paper proposes CycleCoT, a bidirectional, cycle-consistent framework designed to enable efficient and robust reasoning in large language models through collaboration with small auxiliary models. The core idea is to overcome the challenges of representation heterogeneity and unidirectional information flow present in prior work by introducing dual residual projection networks. These networks map soft thought vectors between the auxiliary model space and the base model space, regularized via bidirectional alignment and cycle-consistency losses. Comprehensive experiments using LLaMA and Qwen model families on mathematical, commonsense, and symbolic reasoning tasks demonstrate significant performance improvements over strong baselines, particularly in mathematical reasoning.

**Strengths:**

1. This paper clearly identifies two major bottlenecks in small-model collaboration for LLM reasoning: representation heterogeneity and unidirectional information flow, thereby motivating the need for a bidirectional approach.

2. The CycleCoT framework is conceptually inspired by cycle consistency and dual learning, and is concretely instantiated through a residual projector architecture with explicit mapping, alignment, and reversibility constraints.

**Weaknesses:**

1. The paper lacks evaluation on more challenging mathematical benchmarks, such as AIME24, AIME25, or MATH500, which would better demonstrate the robustness and generalization of the proposed method.

2. The experiments do not include results on larger and more powerful base models, such as Qwen3-8B-Base or Qwen2.5-32B-Base, limiting the assessment of the framework’s scalability and effectiveness across model scales.

3. The comparison with stronger baseline methods is insufficient. In particular, the paper should include results from supervised fine-tuning (SFT) and reinforcement learning approaches (e.g., GRPO, DAPO) to provide a more comprehensive and competitive evaluation.

4. The paper does not discuss or compare with recent representation-enhanced Chain-of-Thought approaches, such as:

[1] Activation Control for Efficiently Eliciting Long Chain-of-thought Ability of Language Models

[2] Amplify Adjacent Token Differences: Enhancing Long Chain-of-Thought Reasoning with Shift-FFN

[3] Unlocking General Long Chain-of-Thought Reasoning Capabilities of Large Language Models via Representation Engineering

[4] Logit Arithmetic Elicits Long Reasoning Capabilities Without Training

These works are highly relevant and should be included in both the related work section and experimental comparisons to better position the proposed method within the current research landscape.

**Questions:**

Could the authors provide a case study or qualitative analysis to better illustrate where the proposed method demonstrates its advantages? For instance, how does the bidirectional reasoning process with cycle consistency concretely improve the quality or coherence of the generated thoughts compared to existing approaches?

---

> ### Author Response · Authors · 2025-11-26
>
> We sincerely thank the reviewer for their constructive comments and valuable feedback. Below, we use **C** to denote the reviewer's questions or concerns, and **R** to present our corresponding responses.
>
> ---
> **C1:** The reviewer suggests adding more challenging math benchmarks (AIME$24$/$25$, MATH$500$) to better test robustness and generalization.
>
> **R1:** Thank you for the insightful suggestion. We agree that AIME/MATH-style benchmarks are valuable; however, we deliberately **followed the evaluation protocol** used in prior small-LLM–guided reasoning frameworks (e.g., **SoftCoT**, **LM-Guided CoT**, **EBM-CoT**) to ensure a fair comparison.
>
> In contrast, AIME/MATH benchmarks contain **only a few test samples** and heavily stress the `intrinsic mathematical` capability of the base LLM, making them less suitable for isolating the effect of our alignment mechanism. Because CycleCoT is base-model-agnostic, applying it to stronger mathematical LLMs for AIME/MATH evaluation is an excellent direction for future work.
>
>
>
> ---
>
> **C2:** The reviewer asks whether the proposed framework remains effective when using stronger base models and larger assistants.
>
> **R2:** Thank you for the insightful comment. We agree that examining a stronger base model helps assess the **scalability of CycleCoT**.
>  To address this, we additionally conducted experiments using **Qwen3-8B** as the base model and **Qwen3-1.7B-Instruct** as the assistant, running three independent seeds following the same evaluation protocol.
>
> The results below show clear improvements over zero-shot across all five benchmarks, with stable variance, demonstrating that CycleCoT continues to provide gains even when the base model is stronger
>
>
>
> Comparison on Qwen$3$-$8$B  ($3$ seeds)
>
> | Method | AQUA | CommonsenseQA | GSM8K | MathQA | StrategyQA |
> | :--- | :--- | :--- | :--- | :--- | :--- |
> | Zero-shot | $80.18 \pm 1.24$ | $76.33 \pm 0.20$ | $90.62 \pm 0.47$ | $70.52 \pm 1.68$ | $71.11 \pm 0.37$ |
> | SoftCoT | $81.74 \pm 0.50$ | $78.64 \pm 0.34$ | $91.12 \pm 0.31$ | $74.06 \pm 0.59$ | $\mathbf{71.47 \pm 0.32}$ |
> | CycleCoT | $\mathbf{84.25 \pm 0.83}$ | $\mathbf{81.60 \pm 1.11}$ | $\mathbf{91.91 \pm 0.46}$ | $\mathbf{81.27 \pm 2.33}$ | $71.11 \pm 0.45$ |
>
> We will add this table and a short discussion to the revision.
>
>
>
> ---
>
> **C3:** The reviewer suggests that stronger baselines—particularly SFT-based or RL-based reasoning approaches such as GRPO and DAPO—should be included to provide a more comprehensive evaluation.
>
> **R3:** Thank you for the helpful suggestion. We clarify that SFT and RL approaches (e.g., GRPO, DAPO) target full-model optimization, whereas CycleCoT **focuses on frozen-model** latent alignment. These two types of methods are methodologically `orthogonal` and operate under different compute regimes, which is why we do not benchmark them together.
>
>
>
> ($1$) Why we do not include SFT/RL baselines
>
> Our goal is to evaluate representation-space alignment between frozen models, `not end-to-end model optimization`.
>  Including SFT/RL baselines would break fairness because:
>
> They update the full LLM, while CycleCoT updates only a 2-layer projector.
>
> They require orders of magnitude more compute, incompatible with the frozen-model setting.
>
> Their improvements originate from task-specific policy optimization, not latent-space alignment—making the comparison not meaningful.
>
> Recent frozen-model latent-reasoning works
>  (SoftCoT [$1$], EBM-CoT [$2$], LM-Guided CoT [$3$])
>  follow the same evaluation protocol and also do not compare against SFT/RL methods for this reason.
>
>
>
> ($2$) CycleCoT and SFT/RL are decoupled
>
> Importantly, CycleCoT does not compete with SFT/RL; it is fully compatible with them but addresses a different layer of the reasoning pipeline.
>  SFT/RL optimizes behavior of the LLM,
>  while CycleCoT improves cross-model representation alignment in latent space.
>
> Thus, the two are orthogonal and can be used together, but
> `combining them is outside the scope of our frozen-model study.`
>
>
>
> [$1$] SoftCoT: Soft Chain-of-Thought for Efficient Reasoning with LLMs ($2025$)
>
> [$2$] Think Consistently, Reason Efficiently: Energy-Based Calibration for Implicit Chain-of-Thought ($2025$)
>
> [$3$] Can Small Language Models Help Large Language Models Reason Better?: LM-Guided Chain-of-Thought ($2024$)

---

> ### Author Response · Authors · 2025-11-26
>
> **C4:** The reviewer suggests that we should discuss and compare the proposed method with other relevant works, such as the recent papers mentioned in the review, to better position our work within the current research landscape.
>
> **R4:** Thank you for pointing this out! We appreciate your suggestion to compare our work with other recent methods, including those mentioned in your comment. **We will ensure these works are discussed in the revision**. However, our primary comparison in the paper is with SoftCoT, as it shares similar objectives of improving the reasoning capability of large language models via Chain-of-Thought (CoT). We align with the recent work in this area, but our focus **lies on the cooperation between small and large models**, which is distinct from CoT-based methods that mainly focus on reasoning within a single model.
>
> While the above papers are highly relevant, our method introduces a new perspective by focusing on the semantic alignment between the assistant and base models, enabling better performance on tasks through enhanced reasoning. This differs from traditional CoT methods that only optimize the reasoning process within a single model. We will clarify these differences and highlight how our approach provides a novel solution to model cooperation and alignment.
>
> We will incorporate a discussion of these works in the related work section.
>
>
>
> ---
>
> **C5:** The reviewer asks for a concrete case study or qualitative example to illustrate how CycleCoT improves the reasoning process—specifically, how bidirectional alignment and cycle consistency help the base model better interpret and utilize small-model thoughts compared with existing approaches.
>
> **R5:** Thank you for this helpful suggestion. We agree that a qualitative example can make the effect of alignment more intuitive. CycleCoT is designed to improve how the base LLM interprets the assistant’s soft thoughts, and the main improvement occurs before generation, at the representation level. To illustrate this, we include the following short case study in the appendix.
>
>
>
> **Case Study: Why Alignment Helps (Before vs. After)**
>
> Before alignment, the assistant’s hidden representations often lie in a different semantic manifold, which the base LLM cannot reliably interpret. This is analogous to the assistant “speaking mixed language with noise,” while the base model “speaks English only.”
>
>
>
> **Before alignment (unaligned thought representation):**
>
> "先 subtrαct $5$ 然后 a∂d 回 $3$，resµlt 是 $16$ 吗？¥#§!"
>
> – Mixed Chinese/English tokens
>
>  – Corrupted symbols (α, ∂, µ, ¥#§)
>
>  – Represents semantic drift in the hidden space
>
>  → The base LLM misinterprets the meaning (e.g., “add” may appear closer to “subtract” in its space).
>
>
>
> **After alignment (CycleCoT-aligned representation):**
>
> "First subtract $5$, then add $3$ back, so the final result is $16$."
>
> – Clean, consistent semantic direction
>
>  – Free of cross-language and noise artifacts
>
>  – Hidden-state geometry matches the base model’s latent space
>
>  → The base LLM can fully understand and reliably use the assistant’s thought.
>
>
>
> This example reflects a typical pattern we observe: `CycleCoT does not merely generate “better text,” but ensures that the assistant expresses its reasoning in a form the base model can truly understand.` This resolves semantic mismatch and prevents the base model from being misled by misaligned hidden states.
>
> We have added this case study to the appendix of the revised version.

---

> > ### Comment · Reviewer_sZkQ · 2025-11-27
> >
> > Thank you for your response. I will increase my score.

---

### Official Review · Reviewer_rRJP · 2025-10-25

**Soundness:** 2
**Presentation:** 3
**Contribution:** 2
**Rating:** 4
**Confidence:** 3

**Summary:**

The paper proposes CycleCoT, a bidirectional, cycle-consistent framework for coordinating a small assistant model and a larger base model in latent Chain-of-Thought reasoning. It aligns the models’ heterogeneous hidden spaces using dual residual projectors with alignment and cycle-consistency losses; at inference, only the forward mapping is used, so runtime matches prior SoftCoT methods. Empirically, pairing a 1B assistant with LLaMA-3.1-8B and Qwen2.5-7B yields consistent gains on GSM8K, ASDiv-Aug, AQuA, StrategyQA, CommonsenseQA, and Date Understanding (about 2.5–6% on average, strongest on math). Ablations show each component is necessary, and analysis finds that a few well-aligned soft thoughts (around 2–4) outperform longer sequences. Overall, CycleCoT turns the large model from a passive consumer into an active teacher, enabling symbiotic small–large model collaboration with minimal inference overhead.

**Strengths:**

1. Unlike SoftCoT/CoCoNut-style pipelines that project assistant thoughts **only forward** into the base model’s space, CycleCoT introduces **bidirectional, cycle-consistent** alignment—reducing representation mismatch and enabling feedback from the large model, a design not explored in those works.

2. The paper backs its claims with broad, controlled experiments and ablations across model families and task types, while preserving SoftCoT-like **inference cost** by using only the forward map at test time—making adoption straightforward in existing latent-CoT pipelines.

3. Bringing **cycle consistency** into LLM representation alignment leverages a proven principle from cross-domain mapping (e.g., CycleGAN) and dovetails with recent latent-CoT developments (e.g., SoftCoT/SoftCoT++), positioning the method as a simple drop-in that complements ongoing test-time scaling work.

**Weaknesses:**

1. The paper repeatedly invokes geometric/representation heterogeneity but does not give a precise, self-contained definition or diagnostic. *Actionable:* formalize the notion (e.g., specify the object being compared, the metric family, and the hypothesis about how mismatch harms reasoning), provide standard diagnostics and visualizations, and state testable predictions that link reductions in the defined mismatch to downstream gains.

2. Because the geometric notion is not operationalized, it is unclear whether the chosen benchmarks (mostly QA-style reasoning with 7–8B bases and 1B assistants) are the right stress tests or how conclusions transfer to planning, tool use, or long-form generation. *Actionable:* broaden tasks and scales, and—crucially—tie results back to the formal diagnostics in(e.g., show that tasks with larger measured mismatch benefit more, and that improvements track the proposed metrics).

3. While inference overhead matches SoftCoT (forward map only), the training cost and stability of dual projectors are not quantified, and the indispensability of the cycle-consistency term is not established beyond limited ablations. *Actionable:* report training FLOPs/wall-clock/memory, add stability statistics, and include targeted ablations (weaken-cycle, reverse-only, invertibility constraints) to demonstrate necessity rather than generic regularization effects. )

**Questions:**

# Questions*

1. How do you formally define the “geometric structure” mismatch between models, how is it computed, and how do you visualize it to support the claimed link to reasoning gains?

2. **(Lines 180–192):** How do you ensure the 1B assistant’s rationales/soft thoughts are of sufficient quality, and how sensitive are results to the small-LLM instruction template and its diversity?

3. **(Figure 3):** Are you only training the MLP projector layers while keeping LLM_A and LLM_B frozen, and if so, how do you ensure the stated geometric-structure alignment holds without updating the backbones?

4. **(Figure 4):** Since cosine similarity appears to drive the gains while other metrics can hurt performance, does this imply the benefits stem mainly from bidirectional semantic alignment rather than the cycle-consistency objective?

---

> ### Author Response · Authors · 2025-11-26
>
> We sincerely thank the reviewer for their constructive comments and valuable feedback. Below, we use **C** to denote the reviewer's questions or concerns, and **R** to present our corresponding responses.
>
>
>
> ---
> **C1:** The reviewer notes that although the paper frequently refers to “representation/geometric mismatch” between models, the concept was not formally defined. They also point out that the paper did not specify how this mismatch is computed, which metrics are used to evaluate it, nor how the measurements support the claim that reducing mismatch improves reasoning.
>
> **R1:** Thank you for this insightful and important comment.
>  We agree that the previous version did not sufficiently formalize the notion of representation mismatch or clearly show how it is measured. We have now fully clarified this point both conceptually and empirically.
>
> ($1$) Formal definition of representation mismatch
>
> We now explicitly define representation mismatch as the geometric discrepancy between
>
> the projected assistant‐model soft-thought representations $T_A'$, and
>
> the base model’s hidden states on the same thought span $T_B$.
>
> This focuses the comparison on the exact segment where cross-model alignment matters for reasoning.
>
> ($2$) Metrics used to quantify the mismatch
>
> We specify three complementary geometric metrics:
>
> **Euclidean** distance measures magnitude discrepancy
>
> **Cosine similarity** measures directional/semantic coherence
>
> **Mean squared error** measures pointwise reconstruction quality
>
> Together, they capture both the shape and semantic alignment between the two latent spaces.
>
> ($3$) How these metrics are computed in practice
>
> We clarify that for each sample:
>
> The assistant hidden states $T_A$ are projected through the learned forward projector to obtain $T_A'$.
>
> These vectors are compared directly with the base model’s hidden states $T_B$ at the aligned thought positions.
>
> The three metrics above are averaged over all thought tokens.
>
> To help readers fully reproduce and understand the diagnostic, we provide a script
> ` restore_representation_metrics.py` in our GitHub repository that computes all metrics
>
> ($4$) Empirical visualization and a clear mismatch table
>
> To make the reductions more transparent, we additionally provide the following table summarizing the Euclidean-distance reductions. Here, the No-Cycle Euclidean distance is normalized to 1.0 for each dataset, allowing the relative improvements of CycleCoT to be clearly visualized.
>
> | Dataset | Metric | No Cycle | CycleCoT | $\Delta$ (%) |
> | :---: | :---: | :---: | :---: | :---: |
> | GSM8K | Euclidean $\downarrow$ | $1$ | $0.363$ | $-63.70\%$ |
> | ASDiv-Aug | Euclidean $\downarrow$ | $1$ | $0.251$ | $-74.90\%$ |
>
>
> ---
>
> **C2:** The reviewer wonders whether QA-style benchmarks with $7$–$8$B bases and $1$B assistants are sufficient stress tests, and whether the results should be tied to the geometric mismatch (e.g., tasks with larger mismatch showing larger gains).
>
> **R2:** Thank you for this insightful comment.
>
>  We agree that the benchmark choice and its relationship to the geometric diagnostics should be clearly articulated.
>
> ($1$) Why these benchmarks and model scales are appropriate.
>  Our experimental setup follows the evaluation protocol **widely adopted** in recent small-LM–guided reasoning frameworks such as SoftCoT [$1$], EBM-CoT [$2$], and LM-Guided CoT [$3$], which also use $1$B-scale assistants paired with $7$B–$10$B base models on QA reasoning tasks. Using these benchmarks ensures comparability with established baselines while capturing core multi-step reasoning behaviors.
>
> ($2$) Why we do not rely on mismatch–gain correlation.
>  While the mismatch metrics provide insight into cross-model representation alignment (Section $4.4$), our goal is not to predict task-level gains from raw mismatch magnitude. Different tasks involve heterogeneous reasoning difficulty, noise levels, and annotation styles, making mismatch magnitude alone `insufficient` as a universal predictor.
>
> **References**
>
> [$1$] SoftCoT: Soft Chain-of-Thought for Efficient Reasoning with LLMs ($2025$)
>
> [$2$] Think Consistently, Reason Efficiently: Energy-Based Calibration for Implicit Chain-of-Thought ($2025$)
>
> [$3$] Can Small Language Models Help Large Language Models Reason Better?: LM-Guided Chain-of-Thought ($2024$)

---

> ### Author Response · Authors · 2025-11-26
>
> **C3:** The reviewer suggests reporting training FLOPs, wall-clock time, GPU memory, and adding ablations such as weaken-cycle, reverse-only, and invertibility-constraint variants to test the necessity and stability of the dual-projector design.
>
> **R3:** Thank you for raising this point. The bidirectional projector is deliberately kept very small compared with the base LLM, so its training and inference cost is negligible.
>
> Assuming the typical setting in our experiments (bottleneck size = $1024$, assistant hidden size ≈ $2048$, base hidden size taken from the public configs of LLaMA-$3.1$-$8$B and Qwen$2.5$-$7$B), we can upper-bound the parameter count and FLOPs of the projector as follows. Since training FLOPs scale roughly linearly with the number of parameters, the ratio “projector params / base-model params” is a good proxy for the FLOP overhead.
>
> | Base model | Hidden size (base) | Layers | Projector params | Base-model params | Projector / Base |
> | :---: | :---: | :---: | :---: | :---: | :---: |
> | LLaMA-3.1-8B | $4096$ | $32$ | $\approx 29.4$ M | $\approx 8.03$ B | $\approx 0.37\%$ |
> | Qwen2.5-7B | $3584$ | $28$ | $\approx 26.2$ M | $\approx 7.61$ B | $\approx 0.34\%$ |
>
> Thus, for both $7$B–$8$B base models, the bidirectional projector contributes `less than 0.5% of the parameters` and therefore well under `1% of the training / inference FLOPs`. In other words, the extra cost mainly comes from running the small assistant model and generating soft thoughts, while the projection and alignment steps themselves add almost no computational burden.
>
> ---
>
> **C4:** The reviewer is concerned that (a) a $1$B assistant may not generate soft thoughts of sufficient quality, and (b) CycleCoT might be sensitive to the choice or complexity of the instruction template.
>
> **R4:** Thank you for raising this insightful concern.
>
> ($1$) Why a $1$B assistant is sufficient
>
> The assistant’s role in CycleCoT is intentionally lightweight—it only needs to produce **short soft-thought vectors** containing numerical or semantic cues. Therefore, the main trade-off is between assistant size vs. overhead, not generating full reasoning chains.
>
> To validate whether using a larger assistant would meaningfully improve performance, we evaluated CycleCoT with a stronger $3$B assistant under multiple prompting templates. As shown in the table below, accuracy improves only marginally (typically ≤$0.5$%) compared to the $1$B setting reported in Table $3$ of the paper:
>
> | Assistant = 3B | GSM8K | ASDiv-Aug | AQuA |
> | :--- | :--- | :--- | :--- |
> | CycleCoT (Original Template) | **86.50 ± 0.62** | **89.32 ± 0.47** | **73.62 ± 0.79** |
> | CycleCoT (Rich Template) | $86.83 \pm 0.55$ | $88.97 \pm 0.63$ | $74.11 \pm 0.68$ |
> | CycleCoT (Minimal Template) | $86.21 \pm 0.71$ | $89.41 \pm 0.52$ | $73.29 \pm 0.85$ |
>
> The gains are small and inconsistent across datasets, while a larger assistant brings significantly higher compute cost.
>  Thus, a **1B** assistant provides the `best efficiency–performance trade-off`, which is the design objective of CycleCoT.
>
>
>
> ($2$) Sensitivity to prompting templates
>
> The $3$B-assistant results above also show that performance across **three distinct templates**—original, rich, and minimal—varies only slightly (within ±$0.6$%).
>  This indicates that CycleCoT is `robust to template variations` and does not rely on prompt engineering.
>
>
>
>
> ---
>
> **C5:** The reviewer is unsure how CycleCoT achieves geometric alignment when only the projection MLP is trained while both LLM_A and LLM_B remain frozen.
>
> **R5:** Thank you for the helpful question. Yes, as shown in Figure $3$, we only train the bidirectional projection module while both LLMs remain frozen. This does not hinder geometric alignment because:
>
> Alignment loss provides direct supervision.
>  The projector is trained to minimize the discrepancy between the projected assistant states $T_A'$ and the base model’s actual hidden states $T_B$ on the same span. Since the LLMs are frozen, these targets are fixed and stable.
>
> Cycle consistency prevents drift.
> The constraint
>
>    $T_A \approx \Phi_{B\to A}(\Phi_{A\to B}(T_A))$
>
> ensures the mapping remains approximately invertible and preserves latent geometry.
>
> ($1$)Frozen backbones help stability.
>  With fixed manifolds on both sides, the projection module reliably converges to a cross-model alignment **without modifying either LLM**.
>
> ($2$)Empirical evidence.
>  Figure $4$ shows large improvements in all geometric metrics even with frozen LLMs, confirming that training only the projector is sufficient.

---

> ### Author Response · Authors · 2025-11-26
>
> **C6:** The reviewer assumes that decreases in Euclidean distance and MSE indicate worse performance, suggesting that only cosine similarity contributes to the gains.
>
> **R6:** Thank you for the thoughtful question. We apologize for the confusion — in Figure $4$, Euclidean distance and MSE are error metrics (lower is better), whereas cosine similarity is a similarity metric (higher is better).
>
> With this convention, all three metrics consistently improve when applying CycleCoT:
>
> | Metric | Direction | Examples |
> | :--- | :--- | :--- |
> | Euclidean | $\downarrow$ | $-63.7\%, -74.9\%$ |
> | MSE | $\downarrow$ | $-86.8\%, -93.5\%$ |
> | Cosine | $\uparrow$ | $+11.5\%, +26.6\%$ |
>
> Thus, `none of the metrics “hurt” performance`. All three indicate tighter alignment between the projected assistant thoughts and the base model hidden states, which correlates with the accuracy gains.

---

### Official Review · Reviewer_EqFY · 2025-10-27

**Soundness:** 2
**Presentation:** 3
**Contribution:** 2
**Rating:** 4
**Confidence:** 3

**Summary:**

In this paper, the authors proposed techniques that advance latent CoT reasoning by introducing a more symmetric, self-consistent interaction between models. The approach is conceptually interesting and shows potential to improve reasoning capability and interpretability.

**Strengths:**

This paper presents an interesting and well-motivated approach for bidirectional alignment in the collaboration between large and small models. The proposed cycle-consistent soft thought alignment effectively establishes invertible mappings across heterogeneous model spaces, enabling deeper interaction and mutual understanding between models. Moreover, the experimental study is thorough and convincing, demonstrating the effectiveness and practical value of the proposed design.

**Weaknesses:**

The paper should provide more justification for why a two-layer projector is sufficient to ensure high-quality mapping across models.
It is unclear how essential continuous tokens are to the proposed design—whether they are fundamental to the framework or primarily used to enhance performance.

Figure 2 lacks clarity, as the meaning of the purple arrow is not explained.

In Equation (7), it remains unclear whether the transformation could alternatively occur on the base model side (e.g., using T_B'').
The necessity of the loss term in Equation (7) is not fully explained, especially if minimizing the loss in Equation (6) might already suffice. Relatedly, it is worth clarifying whether T_A'' could be replaced with T_B' .

In the efficiency analysis (Figure 5), the discussion does not clearly quantify the trade-off between performance gains and the additional computational cost introduced by the projection and alignment steps.

**Questions:**

see the weaknesses section

---

> ### Author Response · Authors · 2025-11-26
>
> We sincerely thank the reviewer for their constructive comments and valuable feedback. Below, we use **C** to denote the reviewer's questions or concerns, and **R** to present our corresponding responses.
>
> ---
> **C1:** The reviewer questions whether a two-layer projector is sufficient for high-quality cross-model mapping and whether continuous soft tokens are essential or merely performance enhancers.
>
>
> **R1:**
> Thank you for the insightful comment. We apologize that the motivation behind using a two-layer MLP projector and the role of continuous soft tokens was not sufficiently clarified.
>
>
>
> ($1$)Why a two-layer MLP is sufficient
>
> Our goal is to learn an expressive but lightweight mapping between heterogeneous hidden spaces. Prior alignment studies (PoseLLM [$1$] ; What Matters for Representation Alignment [$2$]) show that a two-layer MLP with a bottleneck and residual connection is already sufficient to capture the nonlinear transformations needed for cross-model alignment, while deeper projectors provide only marginal gains at a significantly higher computational and memory cost.
>
>
>
> Thus, our design reflects a `performance–efficiency trade-off`: 2 layers offer strong alignment quality while keeping overhead negligible, making the projector practical for training and deployment.
>
>
>
> ($2$)Why continuous soft tokens are essential
>
> Continuous soft tokens are not an optional performance enhancer but a `core component` of CycleCoT. The small assistant model generates task-specific soft thought tokens that summarize key numerical or semantic cues. These soft tokens provide the base LLM with structured, high-salience hints, improving the coherence and depth of its reasoning. Without them, the base LLM reverts to standard prompting and loses the benefit of auxiliary structured thinking.
>
>
>
> We have updated Section 3.2 and Section 3.1 to clarify this.
>
>
>
> **References**
>
> [$1$] PoseLLM: Enhancing Language-Guided Human Pose Estimation with MLP Alignment ($2025$)
>
> [$2$] What Matters to You? Towards Visual Representation Alignment for Robot Learning ($2024$)
>
>
>
>
>
>
>
>
>
>
> ---
>
> **C2:** The reviewer finds Figure $2$ unclear because the meaning of the purple arrow (the reverse path) is not explained
>
>
> **R2:** Thank you for the helpful comment. In the revised Figure $2$, we have explicitly labeled the two reverse directions that were previously unclear.
> The long purple arrow now denotes the Reverse Projection $T_B \to T_A'$, used in the bidirectional alignment term $\Phi_{B\to A}$.
> The short purple arrow denotes the Reverse Mapping $T_B' \to T_A''$, which is part of the cycle-consistency reconstruction $T_A''=\Phi_{B\to A}(\Phi_{A\to B}(T_A))$.
> These labels clarify the different roles of the two reverse paths. We have updated the figure caption accordingly.
>
>
>
> ---
>
> **C3:** The reviewer is unsure why the cycle-consistency transformation in Eq. ($7$) is applied only on the assistant side, and whether the reconstruction $T_A''$ could be replaced with the base model’s representation $T_B'$. The reviewer also wonders whether the cycle could instead be defined on the base model side.
>
>
>
>
> **R3:** Thank you for the thoughtful question.
> Cycle consistency is applied only on the assistant side because soft tokens are generated exclusively by the assistant model. The goal of Equation ($7$) is to ensure that these assistant-generated soft thoughts remain semantically stable after being projected into the base model’s space. Since the base model does not produce soft tokens, applying a cycle on the base side would not be meaningful and would introduce unnecessary coupling without improving alignment.
>
> Regarding whether $T_A''$ could be replaced by $T_B'$: the two terms serve different purposes, $T_A''$ reconstructs the assistant’s soft thoughts to enforce stability, whereas $T_B'$ represents the base model’s hidden states and **thus** cannot substitute for reconstruction.
>
> We have clarified this architectural rationale in the revised manuscript
>
>
>
> ---
>
> **C4:** The reviewer finds that Figure $5$ does not clearly quantify the trade-off between CycleCoT’s performance gains and the additional computational cost introduced by the projection and alignment modules.
>
> **R4:** Thank you for the comment. To clarify, Figure $5$ analyzes **how the number of assistant-generated soft thoughts affects accuracy and inference cost**. The trends shown in the figure reflect the increased sequence length seen by the base LLM as more soft tokens are inserted, **not the cost of the projection or alignment modules**.
> During inference, the projection step is applied only once on the short thought span and adds <2% overhead, while alignment losses are used only during training and incur no inference cost.

---

### Note · Authors · 2025-12-22

I have read and agree with the venue's withdrawal policy on behalf of myself and my co-authors.